# ‘A Lot of People Just Go for Walks, and Don’t Do Anything Else’: Older Adults in the UK Are Not Aware of the Strength Component Embedded in the Chief Medical Officers’ Physical Activity Guidelines—A Qualitative Study

**DOI:** 10.3390/ijerph191610002

**Published:** 2022-08-13

**Authors:** Ashley Gluchowski, Helena Bilsborough, Jane Mcdermott, Helen Hawley-Hague, Chris Todd

**Affiliations:** 1National Institute for Health and Care Research Applied Research Collaboration Greater Manchester, Division of Nursing Midwifery and Social Work, Faculty of Biology, Medicine and Health, School of Health Sciences, The University of Manchester, Manchester M13 9PL, UK; 2Manchester Academic Health Science Centre, Manchester M13 9PL, UK; 3Manchester Institute for Collaborative Research on Ageing, The University of Manchester, Manchester M13 9PL, UK; 4Manchester University NHS Foundation Trust, Manchester M13 9PL, UK; 5National Institute for Health Research Older People and Frailty Policy Research Unit, Division of Nursing Midwifery and Social Work, Faculty of Biology, Medicine and Health, School of Health Sciences, The University of Manchester, Manchester M13 9PL, UK

**Keywords:** exercise, resistance training, uptake, adherence, policy, recommendations, health promotion, health behavior, ageing, active ageing

## Abstract

Strength recommendations have been embedded within the UK’s Chief Medical Officers’ physical activity guidelines since 2011. In 2019, they were given a more prominent position in the accompanying infographic. However, there is limited evidence that these recommendations have been successful in their population-wide dissemination. This study aimed to explore the engagement of community-dwelling older adults with the guidelines to date and to gain a nuanced understanding of the awareness, knowledge, and action that older adults take to fulfil strength recommendations. A total of fifteen older adults living in the UK participated in one online interview. A general inductive approach was used to generate themes from the data. There were four major themes that were found. 1. The strength component of the physical activity guidelines, 2. Barriers, 3. Motivators, and 4. Solutions. No participants were aware of the strength guidelines. When they were asked what activities they used to fulfil the ‘*build strength on at least two-days-per-week′* criteria, walking, yoga, and Pilates were the most common responses. Ageism and strength training misconceptions were major barriers to participation in strengthening exercise. Older adults were much less aware of the benefits of building strength and strength training participation when compared to aerobic activities, so motivators to participation were generally not specific to strength training. Finally, there are several ways that practitioners can overcome the barriers to strength training participation. Solutions to improving the uptake and adherence to strength training participation are likely to be more successful when they include opportunities for social interaction, ability-appropriate challenge, and provide both short- and long-term benefits.

## 1. Introduction

One in three older adults are classified as inactive in the United Kingdom (UK) [1]. As our population ages, people tend to become less active, unwilling, or unable to perform enough physical activity and structured exercise to meet minimum recommendations [2]. As a result, one in every six deaths are attributed to physical inactivity [2]. It is expected that by the year 2050, 25 per cent of the total population will be over the age of 65 years in the UK [3]. The combination of an inactive and ageing population is already having serious implications on an individual’s health, social and mental wellbeing, disability-free lifespan, and quality of life, as well as placing enormous global burden on entire healthcare systems, pension systems, economic development, and the environment [4].

Muscular strength has been shown to be inversely correlated with multiple age-associated disabilities, diseases, and even all-cause mortality [5,6,7,8,9]. The most efficient and effective way to build muscular strength is through progressive resistance training [10,11]. With an evidence-based, progressive resistance training prescription, adults retain the ability to gain significant muscular strength in later life [12]. In 2020, due to the consistently robust evidence in this area, the World Health Organization (WHO) updated their 2010 report ‘WHO guidelines on physical activity and sedentary behaviour’ to include strength recommendations for older adults. The 2020 report lists a ‘strong recommendation’ for muscle-strengthening activities at a moderate or greater intensity for all major muscle groups on two or more days a week.

Within the UK’s own physical activity guidelines (Chief Medical Officers’ physical activity guidelines, hereafter, CMO PAG), strength recommendations have been included since its inaugural 2011 edition [13]. During its 2019 revision, the strength component encouraging older adults to ‘build strength on at least 2 days a week’ was placed in a more prominent position on the infographic, but otherwise not updated. On the user-facing infographic for adults and older adults (Figure 1), this text guidance is further illustrated with a person with dumbbells in each hand at the gym, a person in a yoga pose, and a person carrying heavy bags [13]. Within the 65-page document, it states that adults (19–64 years of age) should choose activities using ‘major muscle groups in both the upper and lower body and be repeated to failure (i.e., until the muscles feel temporarily ‘tired out’ and unable to repeat the exercise until rested for a short period’ (page 31). Yet, older adults (65 years and over) should ‘maintain or improve their physical functioning by undertaking activities aimed at improving or maintaining muscle, balance, and flexibility on at least two days a week’ [13]. Although exercise intensity of effort guidance is embedded within the WHO strength recommendations for older adults, as well as the aerobic component of the CMO PAG for older adults, it is notably absent within the strength component of the CMO PAG for older adults, despite Public Health England’s endorsement [14].

While the significant and positive benefits of strength training are well known to researchers, population-wide uptake and adherence remains low. It has been reported that 42% of 50–74-year-olds are meeting the strength recommendations (Active Lives Survey November 2020–2021, Sport England). In the United States and Australia, the numbers of older adults participating in recommended resistance training levels are reported in the 1–16% range [15,16,17,18,19,20]. As these figures are self-reported via a questionnaire, the number of people meeting the strength recommendations is likely to be even lower, especially for those who may be unaware of the guidelines or unsure what activities fulfil their requirements. Therefore, the aim of this study is to offer a deeper, more nuanced qualitative description of older adults’ perception of their awareness, understanding, participation in, and adherence to the strength component of the Chief Medical Officers’ physical activity guidelines in the United Kingdom [21].

Research Questions

1.Are older adults aware of the strength recommendations that have been embedded within the Chief Medical Officers’ physical activity guidelines since 2011?2.Do older adults believe they are meeting these strength recommendations? If so, *how?* That is, what activities are they using to meet the strength recommendations?

## 2. Materials and Methods

We took a qualitative approach to the problem, with semi-structured, one-on-one online interviews with a convenience sample that was conducted by the lead author (AG, female, Clinical Exercise Physiologist and researcher with experience and interest in prescribing very heavy load resistance training to older adults for the prevention of age-associated disability and disease, as well as in qualitative interviewing). Qualitative interviews were chosen to allow for social interaction with our participants in order to probe for deep, rich, and meaningful answers, beyond what is typically permitted in a quantitative survey. Furthermore, we may have faced recruitment issues due to the low desirability, willingness, or capacity to type or write out experiences when compared to the ease of having a conversation about them.

Advertisements for the study were included in e-newsletters that were sent from ageing charities in the UK. The inclusion criteria were people that were aged 65 years and over and currently living in the United Kingdom, to align with the age and regional boundaries of the guidelines [13]. The first 17 people who contacted the lead author via e-mail expressing interest *and* who identified as age 65 years or older and currently living in the UK were sent the participant information sheet and consent form. A total of 15 consent forms were returned and subsequently interviewed (11/2021). No reasons were provided for the two consent forms that were not returned.

There was no prior relationship to any of the participants and the participants were not given any knowledge about the lead researcher or reasons for doing the research until after the interview was complete. This was in an attempt to prevent the participants coming to the interview with pre-conceived ideas. The guiding interview questions are shown in Table 1; impromptu follow-up questions allowed the interviewer to further explore the participants’ responses. The interviews lasted approximately 30 min, taking place on, and recorded using the online platform, Zoom. The audio recording was transcribed verbatim. The transcripts were offered to the participants for comment or correction. A total of 5 of the 15 participants took up this offer and only minor corrections were made. No other field notes were used. The transcripts were coded using NVivo 12 (qrsinternational.com). A £25 online gift card was offered as a token of appreciation following the interview.

This study used a general inductive approach to qualitative analysis to describe core narratives or themes, rather than develop theory [22]. The lead author read and re-read each of the transcripts to immerse herself further in the data. Next, codes were applied to segments of the data that included points that were relevant to the study’s aims. The codes were then arranged and rearranged to form building blocks or themes that were found across transcripts [22]. To increase rigor, a second author (HB), independently analyzed a random subset of our data for theme identification, naming, and review. The coders met to discuss the themes and successfully reached a consensus.

All the participants gave their written informed consent prior to any study activities. Informed consent from each participant was verbally confirmed prior to recording the interview. This study was approved by the University of Manchester Research Ethics Committee (2021-12920-20831).

## 3. Results

A total of 15 (7 females, 8 males) participants were interviewed for this study. The participants ranged in age from 65 to 77 years with a mean age of 70 ± 3.3 years. All the participants were living independently in the UK at the time of their interview. All the participants described participating in >150 min of aerobic activity per week and as such, were classified as highly active. Direct quotations are presented to enrich our findings and presented in the text as (Sex, ID Number, Age).

There were four major themes (1. Strength component of the CMO PAG, 2. Barriers, 3. Motivators, and 4. Solutions) and three sub-themes for the first major theme (1a. Awareness and knowledge, 1b. Action, and 1c. Suggestions) and one main sub-theme for the remaining three major themes (2a. Misconceptions, 3a. Benefits, 4a. Social) that were derived from the data. Figure 2 shows the coding tree.

### 3.1. Major Theme 1—Strength Component of the Chief Medical Officers’ Physical Activity Guidelines (CMO PAG)

#### 3.1.1. Main Sub-Theme 1a—Awareness and Knowledge

When asked about the physical activity guidelines, most participants replied that they were aware of the basic aerobic guidelines. There were two participants that correctly described the aerobic guidelines. However, despite claims that they knew about the aerobic guidelines most participants had great difficulty describing them.
*I think it’s something around, you know, making sure that you’ve got a…your active enough to make yourself a bit out of breath and exercise at least thirty minutes, I think. I can’t remember any of the other things, I am sure there are more things in it than that.*(M5, 65)
*I think the advice seems to be to keep moving as much as you can.*(F4, 71)

One of our oldest participants (aged 76) was unable to recall any parts of the guidelines.

None of the participants mentioned the strength component in their initial, unprompted response to the questions asking to provide an explanation of the CMO PAG. Once the CMO PAG infographic was presented and the strength component read aloud, the participants still indicated that they had not heard or seen guidelines relating to strength before the interview.
*I honestly can’t say that I ever recall seeing that.*(M2, 68)
*The answer is probably no. So, no, I hadn’t come across the guidelines.*(F5, 71)

After the strength guidelines were explained, the youngest participant indicated that although he had not heard of the strength component of the CMO PAG, his social circle had mentioned the importance of building strength.
*Well, I’ve not heard of the Government target for that, but I’ve heard people, older people, in particular, talking about that. You know, just to maintain your physical, you know, your skeletal stuff in good order, yes… some kind of weight bearing exercises.*(M5, 65)

A couple of our participants admitted to intentionally ignoring physical activity advice. They believed that because they are doing more (aerobic) activity than the guidelines state or because they believed the guidelines to be the bare minimum, they are irrelevant to their individual circumstances.
*But, no, because I’ve got my own routine, you know, I hear it, I hear it on news programmes. I don’t pay a lot of notice to it because I’m probably doing that and more anyway.*(M8, 67)
*…but it seems the lowest amount of time to me, so I think the Government guidelines are probably the minimum, but I don’t know, I’m not an expert.*(F4, 71)

#### 3.1.2. Main Sub-Theme 1b—Action

We asked the if they believed they were meeting the recommended guidelines of ‘build strength on at least two days a week’ and there was a mixed response. Some were unsure what counted towards the guidelines. Others were sure they were meeting the strength guidelines, but the activities that were most often described were in fact, aerobic. For those who believed they were meeting the strength guidelines, they most often described walking, yoga, and/or Pilates.
*I mean there is a lot of building strength in the [walking] football I do, for instance.*(M7, 66)
*I do a lot of yoga, so it would be standing postures in yoga, walking uphill, bit of running, that kind of thing…*(M4, 69)
*I do, I think I’m more than meeting them because I do resistance training in Pilates, I do masses of walking, which is a muscle strengthening thing.*(F4, 71)
*Certainly, the hydrotherapy session, which I’ve had this morning. The Pilates, yes. The cycling, yes. And the walking, yes.*(M3, 71)

Others were not as confident in describing their current activities as enough to meet the strength recommendations.
*I don’t do anything…other than the running and the swimming, I don’t do anything that’s strength-related, or I guess…so I guess there’s nothing for the upper body. I guess my legs get strong with the running.*(M2, 68)
*I don’t think yoga would because that’s more flexibility, isn’t it, so the answer to that, I think, would be no.*(M6, 68)

Still, others were more certain that they were not meeting the strength component.
*I’m not as disciplined over that, I guess, as I am the other physical activities that I do.*(M5, 65)
*I’m not meeting that, no, no.*(F1, 68)

#### 3.1.3. Main Sub-Theme 1c—Suggestions for Improvements to the Chief Medical Officers’ Physical Activity Guidelines (CMO PAG)

The participants had many suggestions on how the guidelines might be improved. The vague, subjective nature of the strength component (see Figure 1) was said to contribute to misunderstanding of the recommendations. This misunderstanding was corroborated throughout the interviews.
*It’s a bit subjective as to what counts as building strength.*(M7, 66)

The terminology that was used in the CMO PAG was brought up several times. The participants indicated a need for more information before they could understand how to meet the strength recommendations.
*Depends on what you mean by strength.*(M8, 67)

For these seemingly healthy, active participants, guidelines that promote becoming more active or getting stronger because it would ‘benefit health’ or ‘reduce their risk of disease’ were *not* messages that inspired or motivated the participants to meet the guidelines. These longer-term benefits, such as ‘reducing falls’, were felt as irrelevant to this younger, independent cohort. Listing specific, short-term benefits of becoming stronger was seen as preferential to our participants.
*I mean, they’re sort of general, keep active, you know, but really vague. I mean, there’s a lot around prevention of cancer, and I suppose quite a lot about prevention of heart disease. And a lot of the exercise is geared towards heart disease. But, it’s more disease specific, you know, shorter goals, shorter term goals, I suppose.*(F3, 71)

The participants also suggested that overall, they had not personally seen evidence of dissemination or prescription of the guidelines.
*GPs could give out the…you know, a more direct message, I just don’t get the feeling that there’s enough. I’m not convinced that as a society we’re putting it in people’s faces enough. If you go to the doctor’s surgery, why is everybody not given a leaflet?*(M6, 68)
*I don’t know how many GPs do exercise on prescription because that’s another good thing they could do more of.*(F7, 70)

The second major theme was around barriers to strength training participation, culminating with a main sub-theme of 2a. Misconceptions to strength training participation.

### 3.2. Major Theme 2—Barriers to Strength Training Participation

A lack of enjoyment was a big barrier to participating in strength training. Interviewees described a bleak picture of them standing alone in a traditional gym environment, lifting a weight, and lowering it back down.
*I tried it once or twice, but, you know, they are just too boring… I just couldn’t do it as much as I need to because it just doesn’t grab me enough. I’m easily bored. It’s got to be faster and the constant activity and decision making. I’d rather do something useful. But just going to a gym and doing this for an hour, what’s the point? It doesn’t get you anywhere, does it? Literally.*(M7, 66)
*I don’t enjoy it. (a) I’ve never done it. (b) I don’t think I would do it properly now. I know there’s trainers there, et cetera, et cetera, et cetera, so I’m sure they could give me the technique, but I don’t know, it’s just, it doesn’t really appeal to me lifting weights.*(M6, 68)

Yet, unsupervised, home-based, isolated strengthening exercises were equally unattractive.
*It’s alright having YouTube videos on but they were...I think everybody in the country started off with that [during the COVID-19 lockdown] and then it wears off, it gets boring. When you’re doing your own rotation, you cheat, you have poor form, you don’t do it. And, again, it doesn’t do anything for me stood in front of a television.*(F1, 68)

The participants noted that aerobic exercise participation was easier because it required less skill and equipment to participate. Strength training was perceived to require exercise knowledge, motivation, and equipment that most of our participants just did not have.
*If you’re at home, there’s always a million other things to do.*(M2, 68)
*Yeah, we have got weights. We don’t have a variety of them though.*(F7, 70)

The traditional gym environment did not appeal to many of our older adults (especially females), acting as a major barrier to strength training participation.
*I don’t want to go to a gym. I’ve never been, but when you look in, you know, they’re all between 20 and 35, you know?*(F3, 71)
*I don’t like the atmosphere and the structure of gyms, you know, that people are sort of making lots of [grunting sound effects]. And the smell sometimes, is disgusting, I can’t bear it. I shouldn’t have said that, should I?*(F4, 71)
*Half the gym are sat there on their phones so you’re not actually working any muscles anyway.*(F1, 68)

Even for those who were fortunate enough to have a gym nearby, there were either no classes offered at all, or no age-or ability-appropriate classes. For one participant particularly, the lack of ability-appropriate classes in her age-friendly region led to visible frustration. She was motivated to participate in strength training but felt let down by the lack of options.
*There’s a big cohort of us that are what you might call recently retired or young old and the provision for us who are fit and active is sadly missing where I am. So, where I am now there are no classes to go to, the nearest one is a Smile class which is seated exercise, and this is my frustration with trying to find something that’s right for me because gyms don’t put on classes for older people and the community classes that are on are always for the older old.*(F1, 68)

Several participants clearly echoed this perceived unmet need. Our seemingly healthy, aerobically active, and independent participants considered themselves an ‘invisible’ population and discussed how community classes were not serving their younger, more-able needs. Our participants felt classes seemed to focus on and be funded for those that were the oldest and those with chronic conditions.
*If you go the gym, you know, you can get all sorts of courses, I understand, for people who are recovering from strokes or have had heart attacks or various kinds of serious life events.*(M5, 65)

Classes for older adults in transition or living with long-term conditions did not seem to meet our participants’ needs as they were not felt to be challenging enough.
*I don’t need to do chair-based exercises or the walks that are for older people, because I can still walk further than that, so they are just not relevant to me at the moment. I’m not really into the older adults stuff yet. I’m pretending I’m not old.*(F7, 70)
*The local Good Neighbours do an exercise class for the elderly, but this is for housebound, you know, they sit on a chair and wave their arms up and down, and I’m not there. I think there’s something about lumping old people, you know, the over 70s, and we’re hugely diverse, just as the under 70s are, you know?*(F3, 71)

Interestingly, this unmet need and barrier to strength training participation was pointed out by both our youngest and oldest participants. After retirement, our participants felt like they had been forgotten.
*I would just say again this middling thing, about the invisible population because if you’re not on a doctor’s radar, on Social Services’ radar, I could have quite easily retired from work and never crossed my doorstep again because I’m not on anybody’s radar.*(F1, 68)

Adverse events, negative side effects, or injuries were mentioned as a (temporary) barrier to our participants’ aerobic activity. None of our participants reported negative side effects or injuries as a barrier to strength training participation. However, as most were not meeting the strength guidelines, this may not be surprising. A couple of our female participants did mention that their husbands (who were not participants of this study) would not bother to see an exercise specialist for strength training advice. These female participants were worried that their husbands were performing exercises without the correct form, which could eventually accumulate into an injury.
*It’s a bone of contention with me because he won’t have an induction and be told what to do. It’s one of these cases of ‘I haven’t got to my age without knowing what I’m supposed to be doing at a gym.’*(F1, 68)

Going back into an indoor space (such as a gym) while the COVID-19 pandemic was ongoing was revealed to be a real barrier to some of our participants. Other participants were looking forward to increasing their social interaction following the lengthy lockdowns, so were happy to follow the recommended safety precautions (vaccinations/boosters, social distancing, exercising outdoors, mask wearing) in order to resume their group exercise. Some participants said that for them, the benefits from exercise participation (and socialization) had to be weighed against the drawback of being isolated and sedentary due to the fear of infection from SARS-CoV-2.
*I’ve had my vaccinations and the booster. I think it’s a balance, isn’t it? You either do useful exercise and take a bit of a risk or you don’t do exercise and that’s got its own risk as well.*(F6, 77)

#### 3.2.1. Main Sub-Theme 2a—Misconceptions-Ageism

A major barrier to strength training participation was the misconception that later life was automatically and inevitably associated with becoming clinically frail and clinically vulnerable. This association, observed by both younger and older adults alike, can evoke unwarranted pity known as compassionate ageism. Compassionate ageism sees older adults portrayed as a group requiring protection from (in our case) the high intensity of effort that can accompany strength training participation. Generations of hard physical labor may also have contributed to the desire to stop, go ‘*lighter as I get older’*, and to never push oneself too hard. Interestingly, the need to avoid vigorous intensities as one ages was a misconception that was brought up for both aerobic and strength training participation.
*You know, you always know, don’t overload yourself…I never push it [heavy loads on strength training machines]*(M8, 67)
*I’m planning my 80th birthday on the top of the mountain but I realize that if I was going to do that, I’d probably have to give up the running.*(F5, 71)

Again, somewhat paradoxically, our participants seemed to describe strength training as (only) important to participate in when they were older, or if/when they started to see their own body decline. Yet, even when participants had admitted to seeing their body decline, they still only considered strength training to be useful for their future selves.
*I think actually I will plan to pay a little bit more attention to strength as time goes on…when I look at pictures of myself as a college student in the football team and I look at my legs, I can see there’s a lot…like I say to myself I’m the same weight as I was then, but a lot of that weight that was in muscle in my thighs in those days is somewhere else now.*(M4, 69)
*I think if I felt the need, I do have a slightly dicky knee, and I did get some exercises for that, but I’m not terribly good at doing them regularly. And that’s the kind of thing, you know, if I had something wrong with me, then I think I’d be much more likely to do it more, than I am now. I’m waiting for my body to tell me…because I can still walk ten miles.*(F3, 71)

#### 3.2.2. Main Sub-Theme 2a—Misconceptions-Programme

Another major misconception surrounding strength training participation was that the same strength program, at the same intensity of effort, will continue to build muscular strength and provide health benefits for months or even years on end. Increasing the load, volume, intensity of effort, or other methods of exercise progression were never discussed.
*I have been doing [Pilates] intermittently for maybe the last five years but, yeah, I’m now doing at least...I’m doing one class a week and I do a basic set of 20 min or so of exercises most mornings which is very much around stretching as well as some strength stuff.*(F5, 71)
*That’s the same one [video], it’s a fixed one. I’m happy with that one because you’re doing core work as well as leg work, and that sort of thing.*(M1, 76)

Although our participants did not mention that progression was important for strength improvement and maintenance, older adults did realize the importance of progressive techniques in relation to keeping strength classes new and exciting for participants. One of our participants suggested that exercise specialists that were *not* utilizing a variety of exercises or increasing the intensity of effort (examples of progression) in their program, was a barrier to their participation.
*There is an instructor at the gym who does the same routine every Friday and he’s done it for the last 30 years, and I don’t go, it’s boring. Yeah, I think the guy that hasn’t changed his routine, because you know what’s coming up you pace yourself and you’re not challenged. And I think oh, this is boring.*(M1, 76)

One grave misconception and barrier to strength training participation was that health status was down to luck (and socioeconomic status), and that adhering to the strength guidelines would yield no further improvements or benefits to health.
*I’m kind of middle class, you know, and I’ve been very lucky, generally, with my health. And so I don’t know that I actually take very much notice [if there were any benefits to be had from strength training participation].*(F3, 71)

The third major theme was around the motivators to strength training participation, with a main sub-theme around the specific benefits of increasing muscular strength.

### 3.3. Major Theme 3—Motivators to Strength Training Participation

#### 3.3.1. Main Sub-Theme 3a—Benefits of Strength and Strength Training Participation

When asked about the benefits of performing strengthening exercise, the participants knew and spoke of the benefits of their aerobic activity, but rarely did they speak specifically about the benefits of improving their strength. On one of these rare occasions, one participant not currently participating in his in-person Pilates class (due to COVID-19 pandemic restrictions), mentioned that Pilates had helped with strengthening his core, relieving his lower back pain.
*I mean the one thing that I miss currently…I’ve been doing Pilates as well for quite a few years and I like a fairly challenging Pilates class. This lower back problem that I’ve got, Pilates helps with the core. And there are no Pilates classes at the gym at the moment. And I would dearly love to… And quite a number of other people would as well.*(M1, 76)

One participant admitted he was likely not doing *enough* to meet the strength recommendations, despite the added incentive that improving his strength would allow him to avoid (or at least delay) surgery.
*I can probably make the hip muscle stronger to delay any operation on that hip. Same with my knees, you know, because they’re both arthritic, just keep the muscles there as strong as possible. I think if I can strengthen muscles around that it’s either going to relieve the pain or elongate the time until I might need some surgery intervention.*(M8, 67)

The last couple of participants who spoke of the benefits of strength and strength training, citied osteoporosis. Although, again, knowing this benefit did not seem to incite the motivation to perform strength training or adhere to the strength guidelines.
*I guess at my age with potential for bone loss, that might be an issue where I’m not meeting guidelines, but since I haven’t looked at them, I couldn’t really tell you.*(M2, 68)

#### 3.3.2. Minor Sub-Themes in the Major Theme 3—Motivators to Strength Training Participation

Ease of Access or Simply being in Proximity to a Gym or Class Acted as an Enabler for Participation.
*I can’t see any barriers. And I mean the one thing is that although I’ve been using the same gym for so many years, when we moved house 15, 16 years ago it happened that we moved nearer to the gym. And no excuse for not going to the gym, it’s 100 yards away.*(M1, 76)

Fulfilling physical activity experiences in childhood fostered physical activity and exercise participation in later life.
*I’ve always been into sport and into team sport. And I wasn’t very good at it. It wasn’t because I could shine at it. It was certainly more for the team thing, the camaraderie and that kept me going. And I think kept a lot of my contemporaries going as well. I would say it’s absolutely essential.*(M1, 76)

On the other hand, we also interviewed participants who were not particularly active during their childhood or during their working lives. Rather, some found the onset of retirement influential in the uptake of physical activity and exercise in later life.
*I couldn’t say I was ever active as a child…I probably do more now that I’m retired…*(F2, 69)

Finally, social interaction was a valuable and robust motivator to participants. The participants admitted to going through the motions when they were training alone or missing their training session all together without social interaction/accountability to keep them honest.
*I’m not self-motivated enough to keep doing it myself. And I’ve always said if I go to a gym I need to go to a class because I can’t walk out of a class.*(F1, 68)
*I really can see the benefit of a class, that you don’t want to let people down, so you do turn up and it’s not so easy to find excuses for not doing it.*(F5, 71)

The fourth major theme was around the solutions to increasing strength training participation, with a main sub-theme of social interactions.

### 3.4. Major Theme 4—Solutions to Increasing Strength Training Participation

#### 3.4.1. Main Sub-Theme 4a—Social Interactions

A social theme appeared in both motivators and solutions, indicating its importance to increasing strength training participation amongst our older adults. The participants alluded to greater adherence–*if* we were able to make our strength training programs more sociable.
*I can see for people in general looking at the patterns of gym attendance, having a café and getting to know some people, I think there is a social side to exercise and that would come into it…*(M4, 69)

#### 3.4.2. Minor Sub-Themes in Major Theme 4—Solutions to increasing Strength Training Participation

It is evident that we need to do a better job at educating older adults on the activities that count toward the ‘build strength on at least 2 days a week’ guideline. When asked which of their activities were building strength, aerobically-dominant physical activities were often counted towards meeting (and even reported as exceeding) the two-day-a-week strength training recommendation.
*So, all those pool-based exercises and the Pilates, are based around that. So, it’s about strengthening, to try and maintain my standing strength, my balance. And also, and I think I’m not doing as much as I should be doing on my cardiovascular fitness. And I think that’s probably the only deficit one that I have.*(M3, 71)

There is also a need to educate older adults on the short- and long-term benefits of strength training and becoming stronger. One participant mentioned that he would do ‘*whatever*’ exercise he needed to achieve longevity.
*You know, so, (a) if I could live to be 110 sort of thing, that would do. If me doing whatever I’m doing as long as I can helps with all that other stuff that helps me keep me alive and fit and healthy, then I’ll keep doing it.*(M6, 68)

Finally, we need to ensure that we reach older adults via channels that are applicable to them, to disseminate and discuss the strength component of the CMO PAG, and to provide age- and ability-appropriate options that effectively meet the strength guidelines. When we do reach older adults, we need to make sure any required bookings are user-friendly.
*I just tried this week and you’ve got to book online and, you know, I’m reasonably tech savvy but it just was unbelievably complicated to do.*(F6, 77)

## 4. Discussion

The seemingly healthy, community-dwelling, aerobically active older adult participants in this study were unaware that there were governmental guidelines for strengthening exercise. This corroborates the exploratory findings in people with long-term conditions living in the UK [23]. Once we presented the CMO PAG infographic and read the accompanying text, it was clear that the participants did not fully comprehend the strength recommendations based on this information alone. From the interviews it was evident that the participants were not clear on which type of activities build muscular strength. We found that participants frequently counted their leisurely walking activities towards the twice-per-week strength guideline. However, even brisk walking in deconditioned older adults produces only minimal, short-term strength gains [24,25]. Thus, the idea that walking builds strength and contributes to meeting the strength guidelines in community-dwelling older adults is a misconception that needs to be dispelled [26,27].

In addition to walking, yoga and Pilates were also counted towards the strength guidelines. Yoga and Pilates practice can take many forms and is most often practiced on a mat with little-to-no equipment. Both yoga and Pilates can even take the form of simple breathing or stretching exercises, as some of our participants acknowledged [28]. Furthermore, some of our participants explained that their in-person class had been replaced with online practice, making it likely that our participants were not receiving personalized, progressive programming. Indeed, research shows conflicting results on the ability for community-dwelling older adults to gain strength from these types of exercise modalities [29,30,31,32]. Importantly, older adults who do show a measurable increase in muscular strength in short-term studies (due to the novel stimulus to the musculoskeletal system), will eventually see a plateau in strength gain with habitual, unvarying practice in as *early* as four to six weeks [33]. As a result, it has been argued that these activities should be removed from the strength guidelines [27].

Some of our participants did describe activities more closely resembling traditional strength training, yet upon further inquiry, they admitted to not doing this regularly, nor with much intensity of effort or volume [34]. To illustrate, 17% of Australian adults initially reported to participate in strength training, yet when probed further about their intensity, less than two per cent were found to be meeting evidence-based recommendations [15,16]. Even older adults performing strength training under the supervision of a personal trainer subjectively described little improvement in their muscular strength [35]. In support of this subjective perception, the authors were able to subsequently and significantly improve their participants’ muscular strength in just eight weeks, verifying that their ‘resistance-trained’ older adults’ strength was nowhere near optimal [12,35,36].

It is evident that the participants in this study were unable to describe the intended intensity of effort, volume, duration, frequency, modalities, or progression principles that were required to improve muscular strength in later life [4,10]. Thus, we would advise any self-report or population survey aiming to investigate strength training participation or adherence in older adults to probe further and interpret their results with caution. Moreover, participation or adherence to strength guidelines will not necessarily be indicative of an individual or population with optimal, or even adequate levels of strength [36]. Ideally, repeated, objective measurements should be used alongside self-reports, strength programming, and education to ensure older adults build muscular strength well beyond the four-to-six weeks following program initiation.

### 4.1. Suggestions for Improvements to the United Kingdom’s Chief Medical Officers’ Physical Activity Guidelines

Our participants have suggested that the strength guidelines themselves may need to pay more attention to the specific needs of diverse cohorts of older adults to be more effective. For example, our participants suggested that targeted, specific messages to older adults who consider themselves as active would increase their motivation to engage and meet the guidelines [37]. Thus, we have added to the growing number of reports suggesting that a lack of detailed guidance on exercise guidelines is a major barrier for older adults [23,38,39,40,41]. The strength component that is embedded within the CMO PAG infographic does not (yet) display important variables such as intensity of effort and session duration as it does for its well-known, and better adhered to aerobic guidelines [13]. Not performing an evidence-based dose of strength training will result in an absence of substantial benefit, another barrier to strength training participation and adherence [38]. If older adults are not able to benefit from their strength training, they lose their (subjective and objective) ‘fitness,’ and subsequently become less willing and less able to participate in resistance training at intensities or durations that are required to maintain their strength and health [5,38,42]. Thus, we would like to see the next set of older adult guidelines take more of a co-production approach to better meet the needs of diverse cohorts [43].

On the other hand, researchers acknowledge that awareness or knowledge of guidelines alone do not necessarily translate to behavior change [44]. Over 70 per cent of adults who were made aware of physical activity guidelines ultimately made no change to their physical activity levels [44]. However, adults were also two times more likely to report an increase in their activity levels when they heard about activity guidelines from their healthcare professional or from more than one source [44]. Yet, just one of our 15 participants could recall seeing the infographic in a public space, not nearly enough to see/hear the guidelines from more than one source. Indeed, the older adults that were interviewed herein insisted that they had no discussions with their health or exercise professional regarding the importance or benefits of strength or strength training [45,46]. Although some participants did confirm that this conversation would likely improve their uptake of strength training [45,46]. Notably, our seemingly healthy, community-dwelling older adults reported significantly less contact with healthcare professionals or exercise specialists overall, an important consideration for future CMO PAG dissemination strategies. Yet, increasing messaging sources, especially from health and exercise professionals, is an easy way to increase the perceived societal norm of strength training, a concept that is known to increase behavior change [47].

Furthermore, healthcare professionals, exercise professionals, and older adults all need distinct information within guidelines in order to promote, refer, prescribe, or carry out evidence-based strength training. The health care professional should be able to sufficiently explain the importance of strength and strength training. Ideally, they will also be able to direct their patient to a qualified clinical exercise physiologist, one with specific expertise that is matched to the ability of their patient [48]. However previous research confirms a lack of knowledge and confidence in this space to be a major barrier in doing so [46]. If a healthcare provider cannot find the time to discuss preventative strength training, allied health professionals have previously noted that incorporating prevention into their business model has led to a surge in business development opportunities [49]. The exercise provider may require much more information than what is currently available within the CMO PAG document. Although the exercise provider may rely on their previous training or qualifications, the evidence they draw on is not necessarily up-to-date or evidence-based (manuscript in preparation).

### 4.2. Motivators and Barriers to Strength Training Participation

Our results agree with a systematic review indicating that the availability of organized exercise is a major motivator to participation in strength training in older adults [50]. Alongside the availability of facilities, the comfort of facilities has also been mentioned to be a motivator to participation [41,51]. Yet, social benefits were listed as one of the most effective motivators to physical activity participation but also to strength training specifically, which practitioners can use to their advantage [50,52,53]. In agreement with Beauchamp et al. [54], our participants indicated a preference for age- and ability-matching of class attendees when choosing to participate in a strength class [50]. Age- and ability-matching further confirms that a lack of personalization in a strength training program is a barrier to adherence [50].

A couple of our participants revealed that although they had never tried traditional resistance training, they were sure that they would not enjoy it. Previous studies have found that older adults initially resist the idea of resistance training but with initiation, recount a *‘(pleasant) surprise’* and even become self-prescribed long-term adherers after experiencing this modality [35,55]. Older adults who have had the chance to train with evidence-based doses, report other forms of strengthening exercise as *‘slow, costly, and inadequate’* [55]. Indeed, training without an evidence-based prescription (that is, at intensities that are lower than optimal for an individual’s ability level), has led older adults to become dissatisfied with their training, likely perpetuating the narrative that strength training must always be an unenjoyable, boring, and useless form of exercise [56,57].

From our interviews, it was clear that the intensity of effort was often left to the older adults themselves (during both home-based and community-based training). Along with our participants’ accounts, it is known that older adults typically choose low effort intensities, and while in the community, older adults often find themselves with over-cautious exercise specialists [35,56]. A recent review confirms that when adults are permitted to self-select their resistance training load, they select a load that is too light to effectively build muscular strength [58]. Further, as adults age there is a tendency to self-select lighter and lighter loads [58]. Again, we know that exercise programs that are perceived as being ‘too easy’ act as a deterrent to strength training adherence [51,57,59]. Moreover, other benefits of resistance training (for example, increasing bone mineral density and functional capacity), likely require even higher intensities than the intensities that are known to build muscular strength [60,61,62]. Taken together, this evidence supports incorporating *‘moderate or higher’* intensity of effort guidance into resistance training programs, as this has been shown to lead to more efficient results, greater satisfaction, and better long-term adherence in our older adult population [35,51,55,63].

Finally, we have seen that seemingly healthy, active, more-able, and digitally literate older adults may have different motivators and barriers when compared to inactive, less-able, or older adults in poorer health or from disadvantaged backgrounds [50,64,65]. To further illustrate this point, our participants mentioned that attempting to promote strength training for the prevention of physical deterioration, or to reduce their risk of falling were irrelevant messages to them at this stage. A systematic review in this area on the other hand, has reported that promoting strength training for the prevention of falls to be a major motivator in the less-able and oldest older adult cohort [50]. Once again suggesting that different messaging and variations of the CMO PAG infographic may be needed for the older adult cohort (manuscript in preparation).

### 4.3. Suggestions for Strength Training Programmes in the Future

Past physical activity has been shown to be a consistent correlate of current physical activity and adherence [54,66]. Previous physical activity can be directly related to self-efficacy and positive outcome expectations, demonstrated in psychological theories such as the Theory of Planned Behavior to influence exercise intention [67,68]. Since societal norms and stigma are also important determinants of physical activity, it would seem logical to ensure strength training becomes a normal and enjoyable occurrence in the lives of adults early in the life course [18,28,38,66,69].

Social support and accountability seem to be especially relevant to older adults [35,38,52,70]. In fact, strength training adherence has been shown to nearly double when it is performed in a group setting compared to an individual setting [70]. So, in addition to top-down public health campaigns from multiple sources, we can make changes from the bottom-up by asking older adults to champion evidence-based programs, helping them to spread their success stories to their friends, families, and communities, and by encouraging them to bring a friend to class [38,41,51]. Older adults are more likely to participate in a new exercise program when a recommendation comes from someone that they know and trust [67,68].

Older adults in a high intensity of effort, ability-appropriate strength class, have greater adherence when compared to those in a lower intensity class [51,52]. This does, however, require expert supervision by appropriately-trained exercise specialists, who need to ensure older adults are adhering to proper exercise technique [18,52,71]. Older adults who feel safe, self-efficacious, and motivated, are more likely to challenge their self-limiting beliefs [35,38,52,71]. A focus on supervised, evidence-based, and progressive resistance training guarantees short-term benefits and reduces the fear and risk of injury, which ultimately increases the likelihood of long-term adherence to strength training [12,35,52].

Our study has limitations. We recruited a small convenience sample (*n* = 15) of homogenous (mostly white, digitally literate, and aerobically active) community-dwelling older adults. Nonetheless, our findings agree with previous studies with this cohort [35,55,63]. The respondents to the study advertisement within e-mail newsletters were clearly engaged with digital media, may have a particular interest in physical activity and exercise, and can be said to be the easiest ‘reachable’ cohort of older adults. It would be easy to conclude then, in this case, that our participants would also be easiest to reach in terms of disseminating the CMO PAG. Since this was not the case, we can be confident that our results are representative of the seemingly healthy, aerobically active, and digitally literate older adult cohort in the UK. Importantly, these engaged, active, and seemingly healthy older adults may quickly transition into less-able older adults if our evidence-based strength messaging and provision fails to target, or be tailored to, their increased needs.

This research was carried out with older adults that were living in the UK and may not represent other geographical areas, although an overlap was found between this and a previous study of resistance-trained older adults living in New Zealand [35].

## 5. Conclusions

We explored older adults’ awareness, knowledge, and action taken towards fulfilling the strength component that is embedded within the Chief Medical Officers’ physical activity guidelines. Although enablers and barriers to physical activity, and to a lesser extent, strengthening exercise are known, this study shows that after a decade of their initial release and two years on from their 2019 update, older adults in the UK continue to be unaware of the strength guidelines and how to meet these recommendations. We have incorporated our participants’ suggestions into tangible directions and solutions for the future design, dissemination, and implementation of strength training guidelines and programs for the older adult. At the very least, we hope that this research continues to raise awareness of these ‘forgotten guidelines’ by stimulating further conversation and research in this area.

## Figures and Tables

**Figure 1 ijerph-19-10002-f001:**
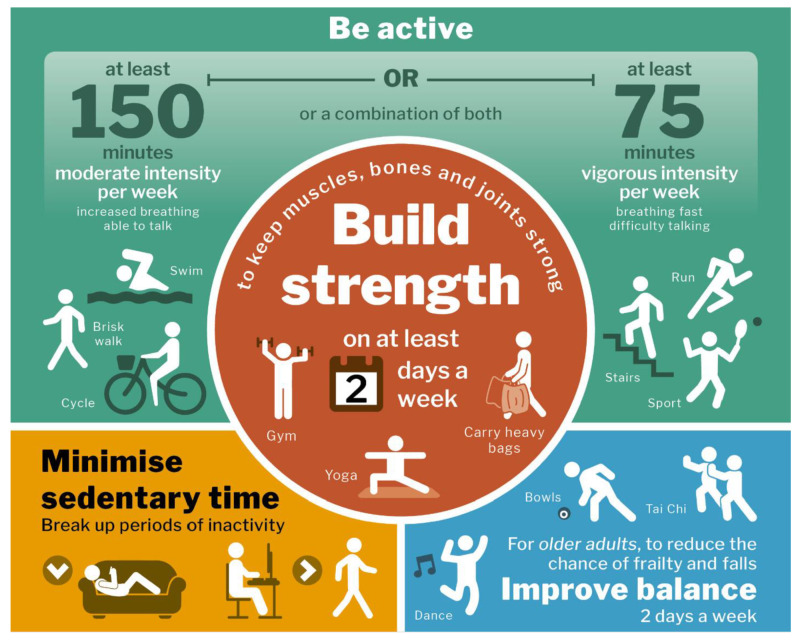
The 2019 United Kingdom’s Chief Medical Officers’ physical activity guidelines infographic (page 35) [13].

**Figure 2 ijerph-19-10002-f002:**
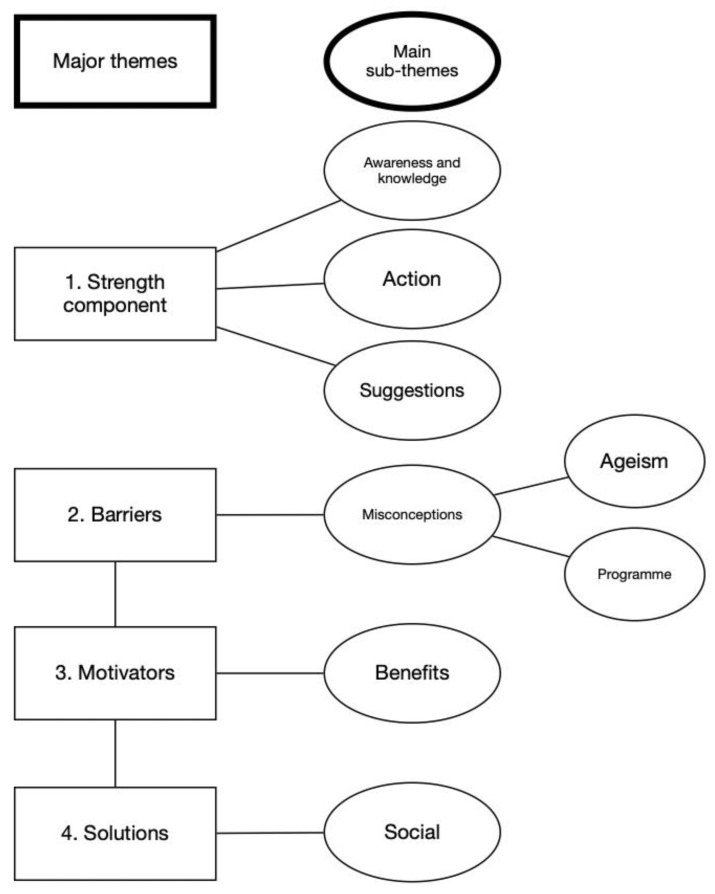
Coding Tree. A total of four major themes were present in the data.

**Table 1 ijerph-19-10002-t001:** Guiding interview questions.

Are you aware of the Chief Medical Officers’ Physical Activity Guidelines? If yes, what are they? If yes, how did you find out about these guidelines? If no, interviewer to show infographic.
2. Do you believe you are meeting these guidelines? If yes, how?
3. What are the benefits of strength training?
4. What are the barriers to strength training? What are the possible negative side-effects?
5. During the COVID-19 pandemic, how do you feel about starting or continuing strength training? Either one-on-one or in a group?

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
