# Peer review of "‘A Lot of People Just Go for Walks, and Don’t Do Anything Else’: Older Adults in the UK Are Not Aware of the Strength Component Embedded in the Chief Medical Officers’ Physical Activity Guidelines—A Qualitative Study"

_ijerph, 2022, doi:10.3390/ijerph191610002_

Round 1

Reviewer 1 Report

I am of the participants' age group in this study, and the findings match everything that I know of my age group in exercise, fitness, and wellness.  Obviously, that comment has to do with the competency of the findings. Older adults don't know and seldom follow the guidelines if they do know them.

The article is well written, easy to read.  Because it is a qualitative study with a very low participant number, the findings are limited.  However, the authors note that limitation.  I would recommend a general survey of some sort to ferret out more quantitative data... not to verify these findings but to convince the agency which developed the strength codes that more is needed:  probably better education from physicians and better education for fitness providers. 

I really liked this paper and found no major or minor errors throughout.  Unfortunately, my age group does not understand the importance of caring for the physical body and most fall into the fallacy that age means a limited exercise and strength program. 

This is one of the few papers that I have reviewed in which I gave almost perfect scores.  And, I review quite often.

Author Response

I am of the participants' age group in this study, and the findings match everything that I know of my age group in exercise, fitness, and wellness.

Obviously, that comment has to do with the competency of the findings. Older adults don't know and seldom follow the guidelines if they do know them.

-We thank you for sharing that your personal experiences parallel our findings!

The article is well written, easy to read.  Because it is a qualitative study with a very low participant number, the findings are limited.  However, the authors note that limitation.  I would recommend a general survey of some sort to ferret out more quantitative data... not to verify these findings but to convince the agency which developed the strength codes that more is needed:  probably better education from physicians and better education for fitness providers.

-We would agree wholeheartedly, and hope that further research by our lab as well as others can do so 

I really liked this paper and found no major or minor errors throughout.  Unfortunately, my age group does not understand the importance of caring for the physical body and most fall into the fallacy that age means a limited exercise and strength program. 

This is one of the few papers that I have reviewed in which I gave almost perfect scores.  And, I review quite often.

-We are truly appreciative and humbled by your time and effort to review our manuscript with such enthusiasm and interest. Thank you!! 

Reviewer 2 Report

This paper qualitatively explores older people’s knowledge of and engagement with physical activity guidelines in the UK. The research problem and premise are sound and the succinct introduction to the paper clearly articulates the purpose. Given the two central research questions, there may be an argument to suggest a quantitative measure with a far larger sample would have provided a more credible response to the research problem. As a qualitative paper, the authors do enough to convince the reader of a fairly rigorous analysis procedure, however with only 7.5 hours (approx.) of interview data there are wider questions about the rigor of the data and the transferability of findings.

Despite these reservations, I liked the paper and would be in favor of seeing it published because of the important themes it addresses.  I have no qualms at all about the construction of the paper itself, which is well-written, fluid, and accessible. The data are interesting, appropriately presented, and deconstructed well in an evidence-based discussion. I believe the paper, while highly critique-able and perhaps not fully qualifying the suggestions for improving the guidelines, does provide an important kicking-off point for further research, debate, and change.

Author Response

This paper qualitatively explores older people’s knowledge of and engagement with physical activity guidelines in the UK. The research problem and premise are sound and the succinct introduction to the paper clearly articulates the purpose.

-Thank you, we truly appreciate you taking the time to review our manuscript.

Given the two central research questions, there may be an argument to suggest a quantitative measure with a far larger sample would have provided a more credible response to the research problem.

  • Further explanation now in manuscript - 

    'We took a qualitative approach to the problem, with semi-structured, one-on-one online interviews with a convenience sample conducted by the lead author (AG, female, Clinical Exercise Physiologist and researcher with experience and interest in prescribing very heavy load resistance training to older adults for the prevention of age-associated disability and disease, as well as in qualitative interviewing). Qualitative interviews were chosen to allow for social interaction with our participants in order to probe for deep, rich and meaningful answers, beyond what is typically permitted in a quantitative survey. Furthermore, we may have faced recruitment issues due to the low desirability, willingness, or capacity to type or write out experiences when compared to the ease of having a conversation about them.'

As a qualitative paper, the authors do enough to convince the reader of a fairly rigorous analysis procedure, however with only 7.5 hours (approx.) of interview data there are wider questions about the rigor of the data and the transferability of findings.

-We have tried to address this in the limitations and will follow up with additional research in this area. 

Despite these reservations, I liked the paper and would be in favor of seeing it published because of the important themes it addresses.  I have no qualms at all about the construction of the paper itself, which is well-written, fluid, and accessible. The data are interesting, appropriately presented, and deconstructed well in an evidence-based discussion.

-Thank you, again, we truly appreciate your interest in our manuscript.

I believe the paper, while highly critique-able and perhaps not fully qualifying the suggestions for improving the guidelines, does provide an important kicking-off point for further research, debate, and change.

  • Thank you, and as suggested, we have tried to address this here - 
  • 'We have incorporated our participants’ suggestions into tangible directions and solutions for the future design, dissemination, and implementation of strength training guidelines and programmes for the older adult. At the very least, we hope that this research continues to raise awareness of these ‘forgotten guidelines’ by stimulating further conversation and research in this area.'